# Neutralized Dicalcium Phosphate and Hydroxyapatite Biphasic Bioceramics Promote Bone Regeneration in Critical Peri-Implant Bone Defects

**DOI:** 10.3390/ma13040823

**Published:** 2020-02-11

**Authors:** Hao-Hueng Chang, Chun-Liang Yeh, Yin-Lin Wang, Kang-Kuei Fu, Shang-Jye Tsai, Ju-Hsuan Yang, Chun-Pin Lin

**Affiliations:** 1Graduate Institute of Clinical Dentistry, School of Dentistry, National Taiwan University, Taipei 10048, Taiwan; changhh@ntu.edu.tw (H.-H.C.); wil1019@ntu.edu.tw (Y.-L.W.); allenfu@hotmail.com (K.-K.F.); shangjye707@yahoo.com.tw (S.-J.T.); 2Department of Dentistry, National Taiwan University Hospital, Taipei 10048, Taiwan; staryeh0524@gmail.com (C.-L.Y.); D06422001@g.ntu.edu.tw (J.-H.Y.); 3Department of Dentistry, Cardinal Tien Hospital Yonghe Branch, New Taipei City 23445, Taiwan

**Keywords:** animal experiments, biomaterials, bone implant interactions, bone substitutes, guided tissue regeneration, bone regeneration, wound healing

## Abstract

The aim of this study was to evaluate the efficacy of bone regeneration in developed bioceramics composed of dicalcium phosphate and hydroxyapatite (DCP/HA). Critical bony defects were prepared in mandibles of beagles. Defects were grafted using DCP/HA or collagen-enhanced particulate biphasic calcium phosphate (TCP/HA/Col), in addition to a control group without grafting. To assess the efficacy of new bone formation, implant stability quotient (ISQ) values, serial bone labeling, and radiographic and histological percentage of marginal bone coverage (PMBC) were carefully evaluated four, eight, and 12 weeks after surgery. Statistically significant differences among the groups were observed in the histological PMBC after four weeks. The DCP/HA group consistently exhibited significantly higher ISQ values and radiographic and histological PMCB eight and 12 weeks after surgery. At 12 weeks, the histological PMBC of DCP/HA (72.25% ± 2.99%) was higher than that in the TCP/HA/Col (62.61% ± 1.52%) and control groups (30.64% ± 2.57%). After rigorously evaluating the healing of biphasic DCP/HA bioceramics with a critical size peri-implant model with serial bone labeling, we confirmed that neutralized bioceramics exhibiting optimal compression strength and biphasic properties show promising efficacy in fast bone formation and high marginal bone coverage in peri-implant bone defects.

## 1. Introduction

Despite the advancing popularity and development of implant dentistry, installing dental implants in patients presenting with poor bone quality poses a challenge [1]. Calcium phosphate ceramics has been frequently used as a biosynthetic material in clinical dentistry [2,3]. Bioceramics such as α-TCP with dicalcium silicate has been proven to be a promising bone substitute in rabbit tibia defect model and might be considered to be used in the field of maxillofacial surgery [4]. Recently, a novel silicocarnotite scaffold, in the tricalcium phosphate-dicalcium silicate binary system, has been advocated to be applied in bone reconstruction, since they imitate the physicochemical characteristics of bone substitutes. Calcium phosphate ceramics play a role in bone formation and also offer a temporary support through the “creeping-substitution” mechanism [5].

Currently, biphasic calcium phosphate composed of a mixture of tricalcium phosphate and hydroxyapatite. (TCP/HA), in particulate or granular form, is possibly the most frequently applied synthetic material [6,7,8]. The most unique property of HA is a chemical similarity with the mineralized phase of bone. This similarity accounts for its osteoconductive potential and excellent biocompatibility [9]; however, HA is limited in use because of its moderate-to-low solubility in the body. Compared to HA, TCP is highly bioabsorbable and biocompatible. The chemical composition and crystallinity of TCP are similar to those of the mineral phase of bone. The rate of biodegradation is high when compared with HA. β-TCP undergoes reabsorption through dissolution and fragmentation over six weeks to 18 months period [10]. However, β-TCP is not replaced with bone at a one-to-one ratio; in other words, less bone volume is consistently produced than the volume of β-TCP that is reabsorbed [10,11]. Thus, β-TCP has been used clinically as an adjunctive with other, less-resorbable bone graft substitutes.

TCP/HA/collagen (TCP/HA/Col) is a composite bone graft material comprising 12% bovine-derived collagen and 88% ceramic, in the form of HA and β-TCP biphasic ceramic (60% HA and 40% β-TCP) [12]. This material has the potential advantage of long-term osteoconductivity for new bone ongrowth of the HA component, and early release of calcium ions from the β-TCP component [12,13]. Collagen is used as an osteoinductive material because of its osteoconductive properties and its compatibility with osteoconductive carriers such as HA and β-TCP. Adding a collagen matrix to these ceramic materials is a simple strategy for coupling the osteoconductive response of two phases (mineral and organic), as well as the porosity and interconnectivity within the materials, to support and encourage cell proliferation and differentiation. Bone marrow aspirate or autograft can also be added to potentially improve the healing of osteoconductive materials [12,14,15,16]. Although the TCP/HA/Col composite can provide a collagen surface for enabling cell attachment, migration, and proliferation, its poor mechanical strength and poor space maintenance in defects compromise its efficacy in large bone defects [12,17].

Dicalcium phosphate dehydrates (DCPD, also known as brushite)-based biomaterials have been prepared in various physical forms and used for bone regeneration. The development of a DCPD setting cement was first reported by Lemaitre et al. [18]. DCPD cements are generally well tolerated by the bone and soft tissue environment in vivo, such that cement resorption is closely followed by new bone formation. DCPDs are particularly promising because of their high solubility under physiological conditions, which has been observed to produce excellent resorbability and enhanced bone formation in vivo, compared with HA-forming cements [19,20]. Thus, a DCP/DCPD-based cement has generated considerable interest [19,21,22].

Previous research has asserted that DCP is a suitable component for grafting bone [2]. Although some researchers have addressed the osteoinductive potential of DCP in certain in vitro and in vivo tests, applying DCP in bone regeneration might be hampered by its high resorption rate and low pH value [23].

DCP-based biomaterials have been prepared in various physical forms, such as injectable pastes, precast blocks, 3D-printed blocks, and granules, and used for bone regeneration [23,24]. DCP cements can be applied to the surgical site alone or in combination with other materials, such as β-TCP and bioglass granules [6]. In these mixtures, the cement provides the granules with mechanical stability and fast resorption, whereas the slow resorbing granules act as bone anchors that facilitate the formation of mature bone [4,23].

Previous studies have successfully tested DCP biomaterials in the regeneration of bone in animal models at various surgical sites, including bone defects at the distal femoral metaphysis, epiphyses and condyle, tibial condyle, and calvaria, as well as bone augmentation in the craniofacial area [6,23]. In these studies, DCP-based biomaterials exhibited osteoconductive potential as well as exhibiting osteoinductive properties during intramuscular implantation [2,25]. However, the amount of bone formation depends heavily on the implantation site and vascular supply, because an adequate blood supply can increase the rate of cement resorption and facilitate the replacement of new woven bone [23]. DCP/HA exhibits acceptable efficacy in osteoinduction and osteoconduction in vitro and in vivo. The speed of new bone formation depends on the bone graft composition and implantation site [20,26].

Recently, we proposed a pH-neutral calcium phosphate bone substitute comprising DCP and HA. DCP/HA biphasic ceramics exhibited favorable biocompatibility and controllable bioresorption rates. This study is the first to investigate the efficacy of the developed DCP/HA biphasic bioceramics as a bone block for promoting new bone regeneration in critical peri-implant bone defects. The commercial bone graft substitute, TCP/HA/Col block, was used as a control group. We chose TCP/HA/Col as a positive control because TCP/HA/Col block is the most commonly used material with a relatively stable shape in our clinic. We would also like to know whether DCP/HA is better than TCP/HA/Col in the repair of critical bone defects.

## 2. Materials and Methods

### 2.1. Reagents and Implants

HA and monocalcium phosphate monohydrate (MCPM) were purchased from Strem Chemicals Inc. (Newburyport, MA, USA). Acetic acid, sodium hydroxide, and other reagents were purchased from Sigma–Aldrich (St. Louis, MO, USA). Sterile collagen-enhanced particulate biphasic calcium phosphate (β-TCP and HA with collagen, TCP/HA/Col) was purchased from Maxigen Biotech (New Taipei City, Taiwan). The ratio of the ceramic components of TCP/HA/Col was 52.8:35.2:12 (wt.%). In addition, the absorbable membrane employed in guided bone regeneration surgery was FormaAid (Maxigen Biotech, Taipei, Taiwan), which is composed of a fibrous collagen matrix, primarily purified from the Achilles tendon of bovines. The cross-linked collagen fibers can effectively extend absorption time. Finally, 3.5 × 8 mm Ankylos A8 dental implants (Dentsply-Friadent GmbH, Mannheim, German) were employed in the experiment. All dental implants with surface treatment were following gamma-ray sterilization, as prepared for routine clinical application.

### 2.2. Preparation of Neutralized DCP/HA

In our pilot study using MCPM and HA to prepare different ratios of DCPD/HA biphasic bioceramics (90%:10%, 80%:20%, 70%:30%, 60%:40%, and 50%:50%), we determined that the DCPD/HA biphasic bioceramic consisting of 70% DCPD and 30% HA exhibited optimal mechanical and solubility properties (data not shown). Thus, in this study, we used an animal model to evaluate the efficacy of DCPD/HA biphasic bioceramics to promote new bone regeneration in critical peri-implant bone defects. The DCP/HA bone-grafting material group was the most crucial experimental group in this study. HA was prepared by heating it to 1300 °C for 6 h and sieving it to obtain particles exhibiting sizes ranging from 32 to 105 μm. A mixture containing HA and MCPM (Strem Chemicals Inc.) was formed. The molar ratio of HA to MCPM was 1:1. Sterilized injection-grade water was added to the mixture to form a paste. The paste was molded using a silicon mold and hardened to obtain a solid material by incubating it at 37 °C for 6 h and at 60 °C for 18 h. The solid material was then washed in distilled water until its surface pH was higher than 6.0. The solid material was dried at 37 °C for 6 h to obtain a DCPD/HA biphasic ceramic. The DCPD/HA biphasic ceramic was packaged and sterilized using gamma irradiation. This material was composed of 70% DCP and 30% HA, and exhibited a porosity of 41.93%. During the 8-week degradation process for the experimental material, the pH value ranged from 6.5 to 7.5, which differed considerably from that of traditional DCP bone cement, the pH of which can drop to 4.2. Our DCPD/HA biphasic ceramic indeed own a neutral pH property different from other traditional DCP with a lower pH value and poor cell compatibility. Therefore, our neutralized DCPD/HA biphasic ceramic with optimal compression strength was chosen for further in vivo functional evaluation.

Both of the bone grafting materials were prepared to form a 10 × 8 × 4 mm^3^ bone block. Subsequently, a 4 mm central hole was drilled into the block to accommodate the 3.5 mm diameter implant (Figure 1b). Finally, scanning electron microscopy was employed to examine the surface characteristics (Figure 1d,f). Two types of crystal characteristics of the DCPD/HA biphasic ceramics were observed under high magnification, as shown in Figure 1d,f, with more flake-like crystal characteristics in the DCPD phase, and a layer of feather-shaped crystals in the HA phase. This may contribute to the early resorption of the material and a promising healing effect on bone regeneration.

The TCP/HA/Col bone graft material was a composite comprising 12% bovine-derived collagen and 88% ceramic in the form of HA and β-TCP. SEM of the TCP/HA/Col group showed porous ceramic granules exhibiting a rough, microstructured surface entrapped within a fibrillar collagen network (Figure 1c,e). Two types of crystal characteristics of the DCP/HA biphasic ceramics are shown in Figure 1d,f. One characteristic was feather-shaped crystals with a regular shape, and the other was a flake-like structure comprising different dimensions. The DCPD phase mostly exhibited flake-like crystal characteristics, with a layer of feather-shaped ceramics, which was ascribed to the HA phase. Biphasic characteristics were observed in the DCP/HA groups under high magnification, which might have contributed to the early resorption of the material and the promising healing effect on bone regeneration.

### 2.3. Animals

Six male beagles (weight: 8–10 kg) exhibiting intact dentition and healthy periodontium were used in this experiment because the size and morphology of the jaw bone were optimal for implantation using guided bone regeneration. The animal experiments conducted in this study were approved by the Institutional Animal Care and Use Committee (IACUC) of National Taiwan University College of Medicine and College of Public Health. (IACUC Approval No: 20120544)

### 2.4. Implant and Bone Substitute Insertion

#### 2.4.1. Tooth Extraction

The animals were anesthetized using subcutaneous injection with a mixture of 2 mL of Zoletil-50 (0.2 mL/kg; Virbac, Carros, France), 0.2 mL of Rompun (Bayer, Leverkusen, Germany), and 0.5 mL of Atropine (0.05 mL/kg) to reduce vomiting and maintain a regular heart rate. After 5–10 min, the animals were under general anesthesia. Local anesthesia was then administered on the surgical field by using 1.8 mL of Xylestesin-A containing epinephrine (1:100,000; 3M ESPE, Seefeld, Germany). The animals were anesthetized, and the 4 bilateral premolars (P1, P2, P3, and P4) and first molar (M1) were then extracted using elevators and forceps. The wound was sutured using 3-0 absorbable sutures, and the stitches were removed for better hygiene care after 7 days.

#### 2.4.2. Defect Creation, Dental Implant Insertion, and Bone Graft Insertion

After the extraction socket had healed (Figure 2a) for at least 10 weeks, the animals were subjected to a surgically created defect, dental implant insertion, and bone grafting. The implant site operations were performed at 4, 8, and 12 weeks, after which the animals were euthanized. All surgical procedures were performed under general anesthesia.

A mid-crestal incision was made, and full-thickness mucoperiosteal flaps were raised. Rectangular saddle-like through-and-through defects were surgically created using a mini-surgical saw (Figure 2b,c) and abundant irrigation with sterile saline [27,28,29]. Meticulous efforts were expended to standardize the size of the saddle defects (depth: 4 mm; width: 10 mm; Figure 2f). An implant (diameter: 3.3 mm; length: 8.0 mm) was installed in the center of each defect (Figure 2d). The top of each implant was placed level to the crest and inserted into the cancellous bone (sink depth: 4 mm). Under-drilled osteotomies were prepared for implant placement to obtain optimal primary stability [27].

Marginal bone height (MBH) was measured as the distance between the marginal bone crest and the bottom of the dental implant. The marginal bone crest was considered the uppermost point of the marginal bone along with the interface between the implant and bone. Total implant height (TIH) was the total height of the implant from the bottom of the implant to the top of the implant. The percentage of marginal bone coverage was calculated by dividing MBH by TIH to indicate the effective bone coverage over the dental implant.

Alveolar-ridge augmentation was implemented at the saddle defects proximal to the implants in the DCP/HA and TCP/HA/Col groups, as well as in the control group without grafting, based on the random selection of animals to undergo this treatment. Resonance frequency detection for implant stability evaluation was performed by using the Osstell Mentor device (Osstell, Göteborg, Sweden) to measure the implant stability quotient (ISQ) [30]. A torque of 10 N·cm was applied to tighten the smart peg in the implant. ISQ values were set on a scale from 1 to 100. The measurement was performed in buccal–lingual and medial–distal directions, both three times for each sample, and an average value was determined for each implant. Implantation and resonance frequency detection were performed simultaneously to ensure sufficient primary implant stability during surgery (ISQ > 50). A FormaAid absorbable collagen barrier membrane was used to cover the implants and defects, extending 2 mm beyond all surgical margins (Figure 2h). The membranes were secured using retention sutures (4-0 Dexon). Subsequently, the primary wound was closed using a horizontal mattress and interrupted 4-0 nylon sutures (Figure 2i).

### 2.5. Radiographic and Histological Assessment

#### 2.5.1. Radiographic Examination

Implant stability was assessed using a resonance frequency detector at 4, 8, and 12 weeks, after which the animals were euthanized. Subsequently, the new bone formation and osseointegration at the bony defects proximal to the 18 implants were assessed using the ISQ, as well as radiographic and histological analyses, and bone labeling. Clinical radiography was employed to evaluate the degree of effective bone coverage over the dental implant during healing. A section of the implant along the coronal plane was used to create a buccal–lingual sectional plane (Figure 2a) [28,29]. A radiograph was captured for the sectional plane for radiograph assessment. Marginal bone height (MBH) was measured, using a radiograph, as the distance between the marginal crest of the host bone and the bottom of the dental implant. The marginal bone crest was determined based on the peak bone level along with the interface between the implant and bone. Total implant height (TIH) was measured from the bottom of the implant to the top of the implant. The percentage of marginal bone coverage (PMBC) was calculated by dividing the MBH by the TIH (MBH/TIH) to indicate the degree of bone coverage over the dental implant. Finally, the mean of both the buccal and lingual percentages of bone coverage was used for the radiographic and histological healing assessment.

#### 2.5.2. Histological Assessment

Coronary sections were made perpendicular to the anterior-posterior axis of the mandible of the euthanized animal (Figure 2c). Sections of the implant were retrieved from the mandible and fixed in a 4% paraformaldehyde solution. Decalcified and non-decalcified (ground) sections were prepared using previously described methods [31,32]. A histological examination was performed using the Zeiss Primo Star microscope (Zeiss, Heidelberg, Germany) equipped with an iCAM-500 CMU imaging system (CMU, Taipei, Taiwan).

After the tissues were initially fixed, they were fixed further in 10% neutral buffered formalin, decalcified in 5% formic acid, embedded in paraffin, sectioned in 5 μm increments, and stained with hematoxylin and eosin (H&E). Sections of the implants along the coronal plane were used to obtain buccal–lingual sectional planes. As in previous radiographic assessment, MBH was measured as the distance between the marginal bone crest and the bottom of the dental implant. The highest point of the crest bone was defined as the highest point of the bone along with the interface between the implant and bone. TIH was the total height of the implant from the bottom of the implant to the top of the implant. The PMBC was calculated using MBH/TIH, which was used to indicate the effective bone coverage over the dental implant. We use the histopathological examination by qualitative observation on the section of the decalcified specimen with H&E staining. Inflammatory cell identification was then recorded and compared among groups.

### 2.6. Bone Labeling and Fluorescent Microscopy

In this study, we used a calcein green and alizarin complexone injections at 4 and 2 weeks, respectively, before terminating each observation time point. In other words, the specimens that were harvested at the 4-week time point were labeled with calcein green to indicate bone remodeling following the operation (0 week), and labeled with alizarin complexone only at 2 weeks after the bone graft implant. Similarly, the specimens harvested at the 8-week time point were labeled with calcein green to indicate bone remodeling 4 weeks after the operation, and labeled with alizarin complexone 6 weeks after the operation. Thus, the specimens harvested at the 12-week time point were labeled with calcein green to indicate bone remodeling 8 weeks after the operation and labeled with alizarin complexone 10 weeks after the operation.

To evaluate the intensity of fluorescein, we used a semi-quantitative grading system, where 3 indicates diffused intensive bone labeling, 2 indicates localized intensive bone labeling activity, 1 indicates weak bone labeling activity, and 0 indicates no obvious bone labeling activity.

### 2.7. Statistical Analysis

The analysis results for both the continual and categorical variables are presented as the mean and standard deviation (mean ± SD). Differences in the measured properties among the groups were evaluated using a one-way analysis of variance with 95% confidence intervals. A *p*-value of less than 0.05 indicated statistical significance.

## 3. Results

### 3.1. Resonance Frequency Analysis of Implant Stability

At four, eight, and 12 weeks, the DCP/HA-treated defects exhibited the highest ISQ, followed by those treated with TCP/HA/Col and the control group (Figure 3a). The results of the resonance frequency analysis indicated significantly higher implant stability for the DCP/HA-treated defect group than for the control group at 8 and 12 weeks post-implantation (*p* < 0.05).

### 3.2. Radiographic Assessment of Defect Healing

As shown in Figure 3b, at 4 weeks, the DCP/HA- and TCP/HA/Col-treated defects exhibited a higher PMBC with a degree of radiopacity at the implant surface compared with the control group. At 8 and 12 weeks of healing, all of the groups exhibited various degrees of bone healing; however, the highest PMBC in the radiographic evaluation was observed in the DCP/HA-treated defect group, followed by the TCP/HA/Col-treated defect group and control the group. Radiographic images of the dental implant and surrounding bone following cross-section on the buccal–lingual plane, among different groups at different time points, are shown in Appendix A.

### 3.3. Histopathological Assessment of Various Grafts Used in Defect Healing

At four weeks after graft insertion, the sections exhibited significantly increased amounts of new bone formation in the DCP/HA- and TCP/HA/Col-treated defect groups (*p* < 0.01; Figure 3c). At 8 weeks, further new bone formation was observed in the DCP/HA group, whereas significantly less bone formation was observed in the control group. At 4, 8, and 12 weeks, the DCP/HA-treated sockets exhibited the highest PMCB, followed by those treated with TCP/HA/Col, and the control group ground and decalcified sections of the dental implant and surrounding bone.

Histological examination of the healing process at the dental implant site and surrounding bone was performed using both ground and decalcified sections (Figure 4). In the control group, marginal bone loss (MBL) was observed at 4 weeks. Prominent inflammation contributing to MBL was observed at the decalcified sections in the control group. Although MBL was observed, some woven bone formation below the periosteum could be identified at 8 weeks after the operation. 12 weeks after surgery, marked inflammation (arrow in the decalcified section) with severe MBL was still observed.

In the TCP/HA/Col group, inflammation (arrow) around the implant and some remaining grafts were observed at 4 and 8 weeks. At 8 weeks, initial new immature bone formation was noted; however, at 12 weeks, marginal alveolar bone formation was observed, and no discernable residual graft material was found.

In the DCP/HA group, although moderate inflammation was also observed (arrow), prominent woven bone formation (*) was found at 4 weeks. No remaining DCP/HA graft was found. At 8 weeks, the DCP/HA group exhibited much more mature bone formation surrounding the dental implant (*) than the other two groups. In addition, soft tissue inflammation also decreased, with no discernable graft material. At 12 weeks, more woven bone (*) at the top of the marginal bone and a higher level of mature compact lamellar bone (**) formation around the dental implant was observed than in the other two groups.

### 3.4. Bone Labeling for Healing Assessment Using Buccal–Lingual Sections

Three zones were proposed for assessing the healing process on the plane through the buccal–lingual section in the peri-implant defects. In Zone 1, healing occurred at the lowermost portion through the interface between the alveolar bone and implant. In Zone 2, following degradation of the graft or blood clot, subperiosteal healing with new bone formation occurred along the periosteum near the alveolar bone. In Zone 3, healing occurred at the uppermost portion of the new bone, which constituted the upper part of the marginal bone adjacent to the implant.

As mentioned, we used alizarin complexone (at 2 weeks) and calcein green (at 4 weeks) injections for bone labeling to observe the healing process in each group (Figure 5). In the control group, moderate bone loss was observed at the uppermost point of the marginal bone along with the interface of the implant and alveolar bone (yellow arrow). Initiation of the subperiosteal reaction contributed to bone remodeling (red in color at 2 weeks after surgery), which was observed as early as 2 weeks thereafter. At 8 weeks, moderate bone loss was identified in the ground section. However, consistent prominent bone remodeling activity was observed in the subperiosteal region at four and 6 weeks after surgery, with high-intensity calcein green (yellow-green in color at 4 weeks) and alizarin complexone (red in color at 6 weeks) bone labeling. At 12 weeks, marked bone loss was observed in the ground section, and a decline in the intensity of bone labeling at both 8 weeks (yellow-green in color at 4 weeks) and 10 weeks (red in color) indicated that bone remodeling gradually decreased.

In the TCP/HA/Col group, the more marginal bone was maintained than in the control group at four, eight, and 12 weeks. The initial bone remodeling that occurred at the interface between the alveolar bone and implant (yellow-green in color immediately after surgery), as well as the subperiosteal region (red in color 2 weeks after surgery), was substantially similar to that of the control group. Sustained active bone remodeling was observed in the subperiosteal region at four and 6 weeks after surgery with moderate-intensity bone labeling using calcein green (yellow-green in color at 4 weeks) and alizarin complexone (red in color at 6 weeks). The area and intensity of bone labeling gradually increased at 8 weeks (green in color) and 10 weeks (red in color), indicating that subperiosteal bone remodeling was still in progress. Additional woven bone formed after a prolonged subperiosteal reaction. Moreover, weak bone labeling at the upper part of the marginal bone at 8 weeks (green in color) and 10 weeks (red in color) adjacent to the implant was observed (arrow), suggesting that the bone remodeling activity was initiated only at the uppermost point of the marginal bone adjacent to the implant.

In the DCP/HA group following surgery, initial prominent bone remodeling activity was observed near the alveolar bone, implant, and bone graft. The initiation of periosteal reaction contributed to new bone formation, which was also observed 2 weeks thereafter. Strong bone remodeling activity and further woven bone formation were compared with the control and TCP/HA/Col groups as early as the subperiosteal reaction occurred (2 weeks). Diffused high-intensity bone labeling using alizarin complexone and calcein green in the periosteal region was prolonged for 4 weeks after surgery. The intensity of bone labeling was sustained and gradually decreased at six weeks, indicating that the bone remodeling activity had peaked and gradually started decreasing. At 8 and 10 weeks, intensive bone labeling at the upper part of the marginal bone adjacent to the implant was noted. Denser lamellar bone formation was found in the DCP/HA group than in the other two groups.

### 3.5. Chronology of the Healing Activity at Three Zones Based on the Bone Labeling Findings

The three zones of healing activity assessment are illustrated in Figure 6a. Considering DCP/HA as an example, at 4 weeks (Figure 6b), initial prominent bone remodeling activity (yellow-green in color immediately after surgery) was observed near the interface between the alveolar bone and the implant in Zone 1. The initiation of subperiosteal bone remodeling (red in color at 2 weeks after surgery) was observed as early as 2 weeks in Zone 2. At 8 weeks, marked bone remodeling with high-intensity bone labeling using calcein green (yellow-green in color at 4 weeks) and alizarin complexone (red in color at 6 weeks) was observed (Figure 6c). At 12 weeks, strong bone labeling was observed at the uppermost section of the marginal bone (green in color at 8 weeks and red in color at 10 weeks), suggesting that the bone remodeling at the uppermost section of the marginal bone adjacent to the implant was ongoing (Figure 6d). By contrast, diminished subperiosteal bone labeling at 8 and 10 weeks indicated that the activity of subperiosteal bone remodeling had peaked and started to decrease.

The chronology of the healing activity was determined at different zones on the basis of the bone labeling findings. In Zone 1, considerably similar bone remodeling activities were observed among the three groups (Figure 6e). The initial prominent bone activity was observed near the alveolar bone, implant, and bone graft. At 2 weeks, bone remodeling activity decreased and was maintained at a low level during the observation period. In Zone 2 (Figure 6f), bone remodeling activity in the control and DCP/HA groups peaked as early as at 2 weeks. The DCP/HA group subsequently decreased in activity at 8 weeks, earlier than the control group, which decreased gradually at 10 weeks. In Zone 3 (Figure 6g), the DCP/HA group exhibited a higher bone remodeling activity than the TCP/HA/Col group at 8 weeks, whereas no obvious bone remodeling activity was observed in Zone 3 during the entire observation period.

## 4. Discussion

In this study, we developed a neutralized DCP/HA material that exhibits a neutral pH, acceptable degradation rate, and adequate stability. We proposed a specific animal model for evaluating its comparative efficacy in bone regeneration. The results indicated that the efficacy of the new bone formation and osseointegration of the DCP/HA group were superior to those of the TCP/HA/Col and control groups in both quantity and quality of new bone formation. Statistically significant differences among the various groups were observed in the PMBC as early as at 4 weeks after surgery. Statistically significant differences were also found in the ISQ values and the radiographic and the histological PMBC at eight and 12 weeks. Furthermore, we employed a serial bone labeling technique to prove similar patterns of initial healing among the three groups; however, at eight and 12 weeks, the rate of new bone formation in the DCP/HA group was significantly higher than that of the other two groups.

Biphasic calcium phosphate composed of a mixture of TCP/HA in particulate or granular form is possibly the most frequently applied synthetic material. Consequently, chemical and physical properties could compromise the clinical application of TCP/HA in alveolar bone defect regeneration [33,34]. To address inadequate graft strength, numerous studies have proposed combinations of various forms of calcium phosphate [33,35,36,37,38]. In this study, we used TCP/HA/Col (GingivAid, New Taipei, Taiwan) as a graft material to compare with our new DCP/HA graft, because TCP/HA/Col has been proven superior to other currently available synthetic grafts and is a commonly used synthetic grafting material in clinical practices in Taiwan. We; thus, applied TCP/HA/Col similarly to block-type grafts in this study.

Brushite cements are generally well tolerated by bone and soft tissues, and they do not cause long-term inflammation. Following implantation, brushite cements are first enclosed in loose connective tissue [22,39], although they can also be surrounded by fibrous connective tissue if the cement composition is highly acidic [40]. The surfaces of brushite and monetite bioceramics stimulate osteoblast activity in vitro [41]. However, osteoblast differentiation in vivo occurs at some distance from the cement surface [23,39], likely because of its rapid resorption rate in vivo (20 μm·d^−1^), which is similar to the rate of bone growth [33], causing a physical gap between the cement and newly-formed bone during the first week after implantation [23].

Alveolar bone augmentation represents a substantial component of clinical practices aimed at promoting bone formation to fill defects caused by periodontal destruction, tumor excision, or trauma [27,28]. In clinical practice, labial bone defects are a critical concern and difficult to restore. An adequate and reliable method for repairing such peri-implant bone defects is lacking [27]. The canine defect model is a well-established animal model for evaluating guided bone regeneration and the healing of peri-implant defects [29]. Despite ethical concerns and efforts to develop alternatives to animal experimentation, standardized animal models are crucial components in translational sciences and medical technology development [29].

Pharmaceutical animal testing, medical devices, and medical strategies have played key roles in transitioning therapeutics into clinics [29]. Numerous strategies are currently being investigated to address the challenge presented by nonunion; however, adequate testing of such strategies is necessary before they are suitable for human use [29]. Using this model, the most critical condition for bone augmentation in buccal or labial vertical or horizontal bone defects can be evaluated. Some authors have used a similar model with sagittal sections [28,42,43]; however, the results may interfere with adjacent bones. Because bone can heal in a manner that is indistinguishable from the bone that never had a defect, it might not be a supposedly critical bone defect. Thus, we employed a peri-implant model to focus on coronary sections (e.g., buccal–lingual sections) to simulate the most critical clinical status for evaluating the efficacy of the proposed substitute composed of DCP and HA.

The bone proximal to a dental implant plays a crucial role in supporting it [44], and implant stability can affect proximal bone quality [45]. New bone formation and osteointegration following implantation may contribute to dental implant stability, which we assessed using the ISQ value [30,46]. Because resonance frequency analysis with Osstell ISQ measures stiffness, which is a combination of bone–implant contact and bone density around the implant [45], Thus, the efficacy of supporting dental implants by new bone formation and osteointegration induced by the graft can be critically evaluated. Many authors have used the ratio of bone–implant contact to evaluate the bone support around implants. However, some researchers have confirmed that marginal bone is proximal to most stress concentrations by using a finite model [47]. The marginal bone may be the site exhibiting the strongest stress concentration and thus changes in factors such as the quantity and quality of marginal bone may be crucial for achieving implant stability. To focus on the most crucial stress concentration area, we calculated the PMBC rather than bone–implant contact around the implants. Another reason for using PMBC is that the bottom of dental implants proximal to the inferior alveolar nerve can be problematic when using bone–implant contact proximal to the implant for assessing bone regeneration [28].

If the rate of bone healing is high in both quality (e.g., bone density) and quantity, then the supported implant can achieve high primary stability. In our study, a significant PMBC in the DCP/HA group was detected as early as 2 weeks after surgery, which may have been the cause of subsequent significant differences in the ISQ and radiographic and histological PMBC at 8 and 12 weeks.

Bone labeling revealed a similar pattern of healing among the three groups in this study at the initial healing event; however, the rate of new bone formation in the DCP/HA group was significantly higher than that of the other two groups. The defect without a graft exhibited the average speed for healing. Generally, synthetic grafting materials may retard healing because they can cause inflammation and resorption of the grafting material, preventing new bone formation. However, in this study, DCP/HA exhibited a similar healing rate to that of the control group and demonstrated healing with promising bone volume maintenance and increased bone formation. The reason that TCP/HA/Col grafting exhibited compromised results might be that it required additional time for graft resorption, delaying bone regeneration. Conversely, the reason that the proposed DCP/HA synthetic grafting material exhibited promising results could be due to its early resorption, favorable osteoconductivity, neutral pH, and acceptable space maintenance effect.

The key factors for bone regeneration have been proposed as primary wound closure, angiogenesis, the stability of graft, and space maintenance for bone regeneration [29]. In our previous study, we determined that materials exhibiting less inflammation and less tension than the overlying soft tissue may guarantee primary wound closure, materials exhibiting an unstable framework, and high porosity may facilitate angiogenesis, and materials exhibiting sufficient strength may possess favorable properties in space maintenance and stability.

The factors that enable the new DCP/HA biphasic bioceramics to promote bone regeneration are as follows. First, sufficient mechanical strength may maintain the space for bone formation. Our pilot study showed that the compression strength of the DCP/HA block is approximately 15.81 ± 2.77 Mpa, which is much higher than the most frequently used TCP/HA/Col graft (3.54 ± 0.97 Mpa). Second, grafts exhibiting an adequate neutral pH value cause little harm and irritation to intraoral soft tissue. The pH value can be maintained at approximately 6.5 after 12 weeks from the observation period. Oral mucosa overlying the jaw bone is thinner than the soft tissue surrounding the extremities or spinal bone in the body. Not all of the bone graft materials used in orthopedics can be used successfully intraorally. It depends on the potential of irritation and inflammation of the surrounding soft tissues. Bioceramics exhibiting a neutral pH value may cause little soft tissue irritation and inflammation, which can reduce local edematous change (swelling) and tension in the overlying soft tissue. Because primary wound closure of oral soft tissue is a crucial factor for intraoral guided bone regeneration, no soft tissue wound dehiscence or graft exposure was applied to the DCP/HA and TCP/HA/Col groups in this study. Our findings also correspond to previous research on brushite-based cements, which demonstrated that the cement released less H+ and reached pH neutrality earlier is crucial, since cements are more likely to be rejected by the body due to an excessive release of H+ ions [40]. Finally, biphasic crystal characteristics facilitate graft resorption and bone formation. Biphasic characteristics were observed in the DCP/HA groups under high magnification, which may have contributed to the early resorption of material and promising healing effect on bone regeneration. Abundant flake-like crystals characteristic of the DCPD phase, which can be easily resorbed and induce bone formation, in addition to a layer of feather-shaped crystals ascribed to the HA phase, may have acted as a framework for space maintenance and graft stability. Our findings correspond to viewpoints that calcium phosphate ceramics not only play a role in bone formation but also offer temporary support through the mechanism of creeping-substitution [4,5]. Physicochemical characteristics of DCP/HA bioceramics are crucial for efficacy in bone regeneration [48,49,50].

Creeping-substitution was compromised of osteoclastic resorption, creation of new vascular channels, and osteoblastic bone formation, which can be observed in the early process of healing [4,5]. In this study, we used the bone labeling technique to localize the most active zone of bone remodeling at different time frames. We observed the healing process from a bottom-up direction at the buccal–lingual (coronary) section. Initially, following surgery, the most active bone remodeling occurred at the interface of the alveolar bone, dental implant, and graft, which began from the bottom of the defect. Subsequently, the most active bone remodeling was observed at the zone below the periosteum following graft resorption or blood clot organization. Finally, the most active bone remodeling shifted up to the most coronal portion of the marginal bone. The initial and subsequent healing process at the interface between the alveolar bone and graft may attribute to creeping substitution.

## 5. Conclusions

We demonstrated that neutral DCP/HA bioceramics with optimal compression strength and biphasic properties exhibited fast bone formation, high marginal bone coverage, and initial stability of the dental implant in critical peri-implant bone defects. The healing process starts from a bottom-up direction by creeping substitution. Biphasic characteristics of DCP/HA, including early resorption of material, creation of new vascular channels, and osteoblastic bone formation, were observed. The physicochemical characteristics of DCP/HA bioceramics are crucial for promising efficacy in bone regeneration. DCP/HA bioceramics may be an alternative bone graft for the immediate implant and extensive bone defect repair in the future.

## Figures and Tables

**Figure 1 materials-13-00823-f001:**
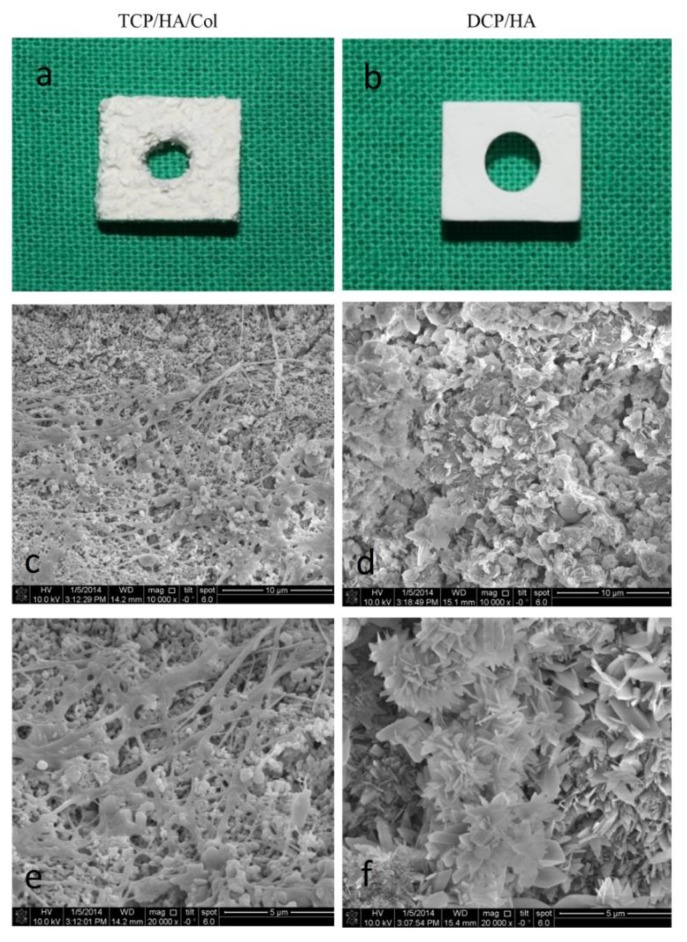
Gross and scanning electron microscopic images of grafts used in this study. (**a**) Gross image of TCP/HA/Col used in this study. A rougher and more irregular surface can be observed for TCP/HA/Col than for DCP/HA. (**b**) Gross image of DCP/HA grafting material used in this study. (**c**–**f**) Microphotographs, obtained using scanning electron microscopy (SEM), showing the surface characteristics of the cross-sections of the TCP/HA/Col (**c**,**e**) and DCP/HA (**d**,**f**) grafting materials. (**c**,**d**): 10,000×; (**e**,**f**): 20,000×.

**Figure 2 materials-13-00823-f002:**
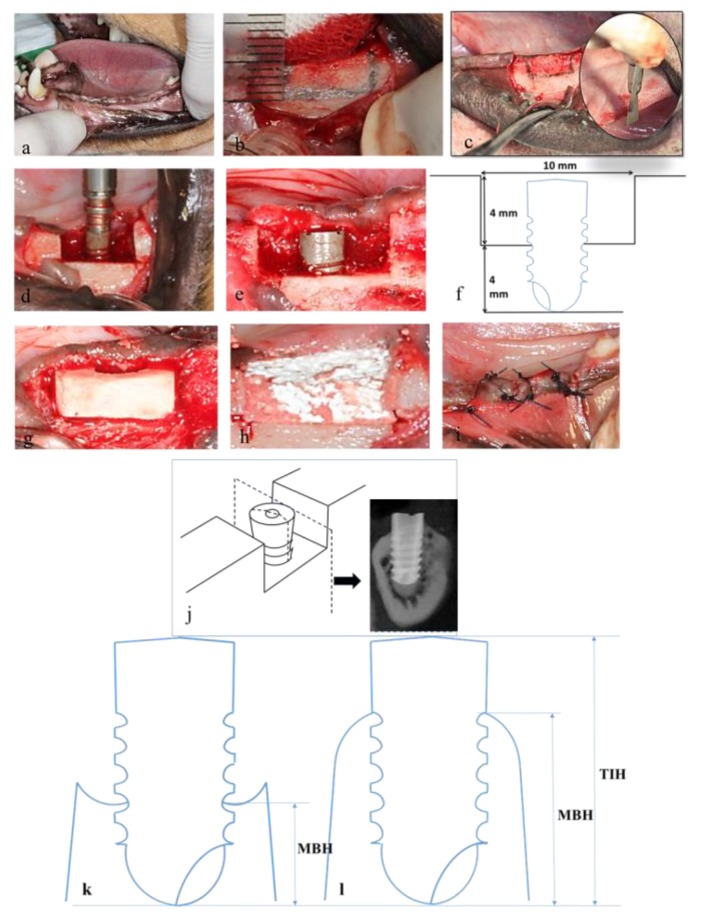
The surgical procedure used for inserting the implant after creating a saddle bone defect and cross-sectional images captured for calculating the final bone coverage. (**a**) Healed edentulous ridge before surgery. (**b**,**c**) Rectangular saddle defects were meticulously marked and created surgically using a mini-surgical saw (circle in (c)). (**d**) Drilled osteotomies were prepared for implant placement. (**e**) The width and depth of the peri-implant bone defect. (**f**) The clinical preparation of an alveolar defect for graft testing. (**g**) Insertion of DCP/HA biphasic ceramics graft. (**h**) Cover the graft with collagen membrane. (**i**) Wound closure with 4-0 nylon sutures. (**j**) The sectional plane cut through the implant along the coronal plane to create a buccal–lingual sectional plane. (**k**,**l**) Cross-sectional images for calculating the final bone coverage when the marginal bone was lost (**k**) or gained (**l**) peri-implant.

**Figure 3 materials-13-00823-f003:**
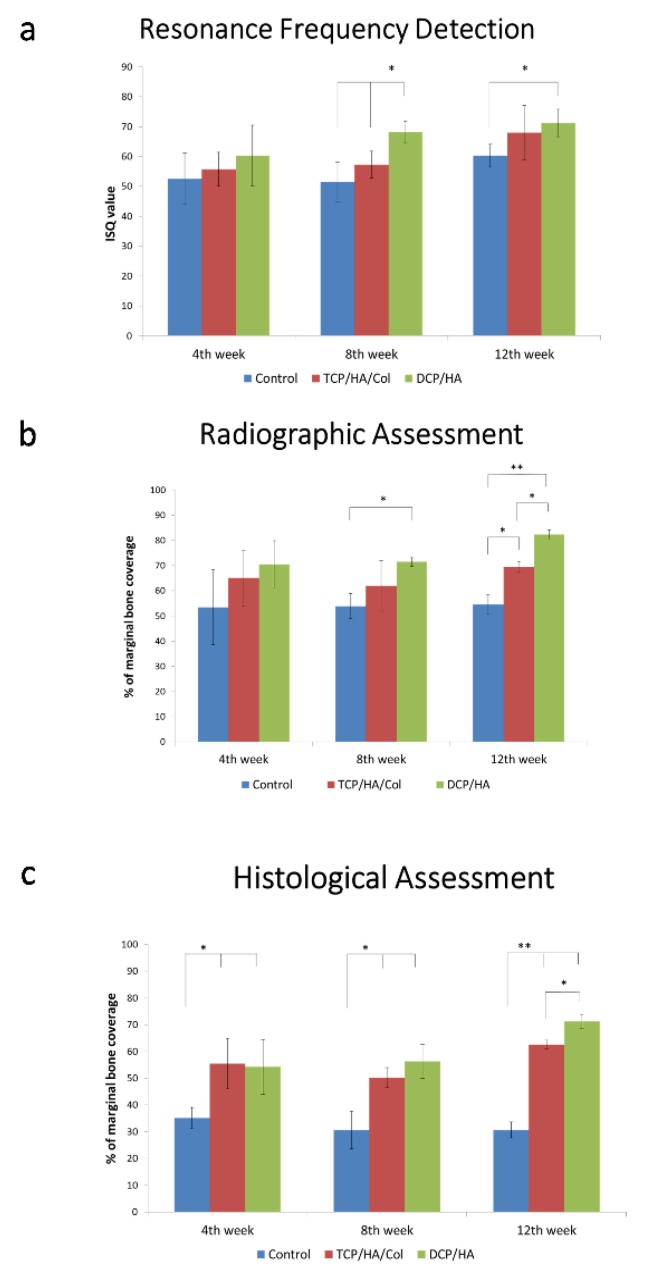
Resonance frequency analysis. Radiographic and histopathological assessment for implant stability after inserting various grafts for healing the peri-implant bony defect. (**a**) After 4, 8, and 12 weeks, the DCP/HA-treated defect exhibited the highest implant stability quotient, followed by the TCP/HA/Col-treated group and the control group. At 8 and 12 weeks following implantation, resonance frequency analysis of the DCP/HA group revealed significantly higher implant stability compared with the control group (*p* < 0.05). (**b**) Radiographic assessment of defect healing. After healing for 4 weeks, the DCP/HA- and TCP/HA/Col-treated defect groups exhibited a higher percentage of marginal bone coverage with a degree of radiopacity at the implant surface than the control group did. At 8 and 12 weeks of healing, all of the groups exhibited various degrees of bone healing. However, the highest percentage of marginal bone coverage based on radiographic evaluation was the DCP/HA-treated defect group, followed by the TCP/HA/Col-treated defect group and the control group. (**c**) Histopathological assessment of the various grafts used in defect healing. At 4 weeks after the graft was inserted, the sections exhibited significantly higher amounts of new bone formation in the DCP/HA- and TCP/HA/Col-treated defect groups (*p* < 0.01). After 8 weeks of healing, an increased amount of new bone formation was visible in the DCP/HA groups, whereas significantly less bone formation was observed in the control group. At 4, 8, and 12 weeks, the DCP/HA-treated sockets exhibited the highest percentage of marginal bone coverage, followed by the TCP/HA/Col-treated sockets and the control group. * and ** indicate statistically significant differences (*p* < 0.05 and *p* < 0.01, respectively) among the labeled groups.

**Figure 4 materials-13-00823-f004:**
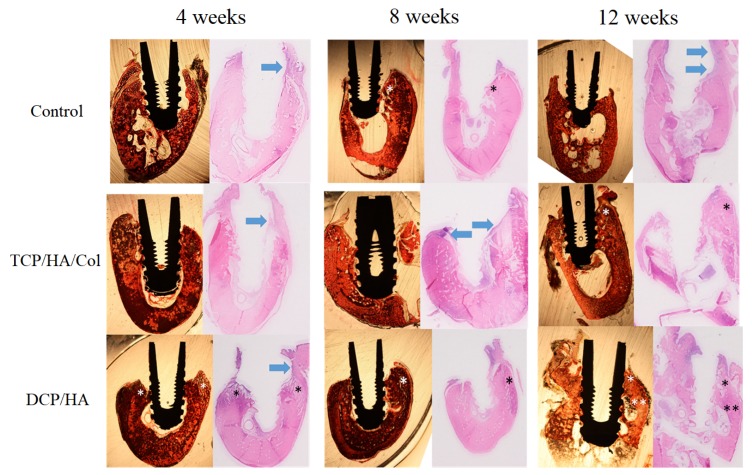
Ground and decalcified sections of the dental implant and surrounding bone. In the control group, marginal bone loss (MBL) was observable at 4 weeks. Prominent inflammation (arrow) that contributed to MBL was observed in the decalcified section in the control group. Although MBL was observed, some woven bone formation below the periosteum was identified 8 weeks after the operation (*). At 12 weeks after surgery, marked inflammation (arrow in decalcified section) causing severe MBL was also observed. In the TCP/HA/Col group, inflammation around the implant and some remaining graft material were observed at 4 and 8 weeks (arrow). At 8 weeks, new immature bone formation was observed. Furthermore, at 12 weeks, more marginal alveolar bone formation (*) was observed than at previous time points. No discernable residual graft material was identified at 12 weeks. In the DCP/HA group, prominent woven bone formation (*) was observed at 4 weeks; however, moderate inflammation was also observed (arrow). DCP/HA graft material could not be easily identified at 4 weeks. At 8 weeks, the DCP/HA group exhibited highly mature bone formation surrounding the dental implant (*), compared with the other two groups. In addition, soft tissue inflammation also decreased without discernable graft material. At 12 weeks, higher levels of formation of woven bone (*) at the top of the marginal bone and mature compact lamellar bone (**) around the dental implant were observed, compared with the other two groups.

**Figure 5 materials-13-00823-f005:**
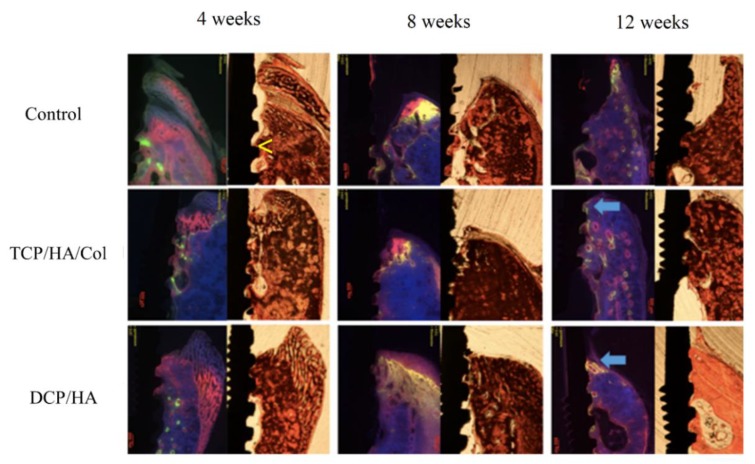
Original ground section and bone labeling indicating the healing process in each group at four, eight, and 12 weeks. At 4 weeks, moderate marginal bone loss (MBL) at the lower uppermost point of the marginal bone along the interface between the implant (yellow arrow) was observed in the control group; however, initially prominent bone remodeling activity (yellow-green in color immediately after surgery) was observed near the interface between the alveolar bone and the implant. The initiation of subperiosteal reaction contributed to bone remodeling (red in color at 2 weeks after surgery) and was observed as early as 2 weeks thereafter. At 8 weeks, moderate MBL was observed in the ground section. However, prominent bone remodeling activity in the subperiosteal region was observed at 4 and 6 weeks after surgery, with high-intensity bone labeling using calcein green (yellow-green in color at 4 weeks) and alizarin complexone (red in color at 6 weeks). At 12 weeks, marked bone loss was observed in the ground section, and a decline in the intensity of bone labeling at both 8 weeks (yellow-green in color at 4 weeks) and 10 weeks (red in color) indicated that bone remodeling had gradually decreased. In the TCP/HA/Col group, the more marginal bone was maintained than in the control group at four, eight, and 12 weeks. The initial bone remodeling that occurred at the interface between the alveolar bone and implant (yellow-green in color immediately after surgery), as well as the subperiosteal region (red in color at 2 weeks after surgery), was substantially similar to that of the control group. Sustained active bone remodeling was observed in the subperiosteal region 4 and 6 weeks after surgery, with moderate-intensity bone labeling using calcein green (yellow-green in color at 4 weeks) and alizarin complexone (red in color at 6 weeks). The area and intensity of bone labeling gradually increased at 8 weeks (green in color) and 10 weeks (red in color), indicating that subperiosteal bone remodeling was still in progress. More woven bone formed after a prolonged subperiosteal reaction. Moreover, weak bone labeling at the top of the marginal bone at 8 weeks (green in color) and 10 weeks (red in color) was observed adjacent to the implant (arrow), suggesting that the bone remodeling activity was initiated only at the uppermost point of the marginal bone adjacent to the implant. In the DCP/HA group following surgery, the initial prominent bone activity was observed near the alveolar bone, implant, and bone graft, initially similar to the control group. Initiation of the periosteal reaction contributing to new bone formation was also observed 2 weeks thereafter. Stronger bone remodeling activity and a greater level of woven bone formation were observed as soon as this subperiosteal reaction occurred (2 weeks), compared with the control and TCP/HA/Col groups. Diffused high-intensity bone labeling using alizarin and calcein green in the periosteal region was prolonged for 4 weeks after surgery. The intensity of bone labeling was sustained and then gradually decreased at 6 weeks, indicating that bone remodeling activity peaked and then gradually decreased. At 8 and 10 weeks, intensive bone labeling at the top of the marginal bone adjacent to the implant was observed to be ongoing. Denser lamellar bone formation was observed in the DCP/HA group than in the other two groups.

**Figure 6 materials-13-00823-f006:**
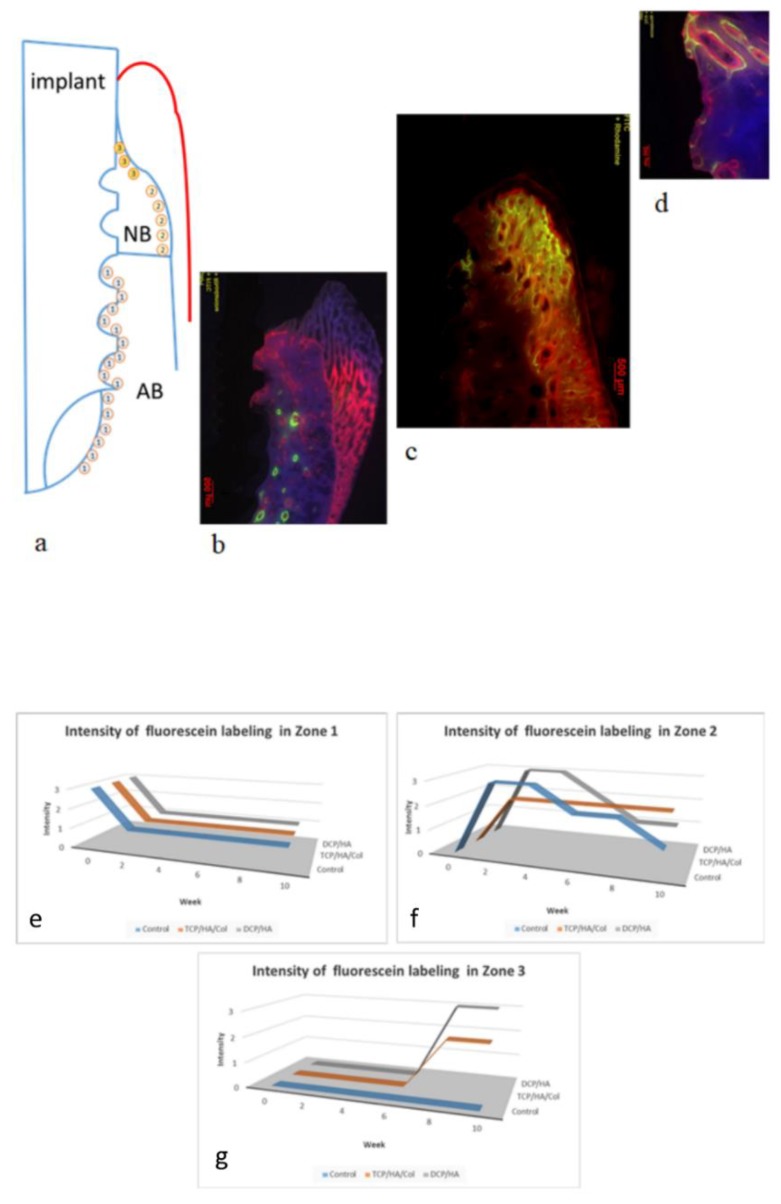
Illustration and bone labeling of the healing process in the buccal–lingual cross-section of a peri-implant bony defect. (**a**) Three zones were proposed for the healing process at different time points. First, healing occurred through the interface between the alveolar bone and implant (Zone 1). Second, subperiosteal healing with new bone formation occurred along the periosteum near the alveolar bone (Zone 2). Finally, healing occurred at the uppermost portion of the new bone (Zone 3). (**b**–**d**) Using the bone labeling image from the DCP/HA group as an example, from the ground section harvested at 4 weeks (**b**), prominent bone remodeling activity (yellow-green in color) was observed immediately after surgery in Zone 1. The initiation of subperiosteal bone remodeling (red in color) as early as 2 weeks was then observed in Zone 2. In the ground section harvested at 8 weeks (c), marked bone remodeling with high-intensity labeling using calcein green (yellow-green in color at 4 weeks) and alizarin complexone (red in color at 6 weeks) was observed in Zone 2. In the ground section harvested at 12 weeks (**d**), strong bone labeling at the uppermost section of the marginal bone (green in color at 8 weeks and red in color at 10 weeks) was observed to be ongoing. By contrast, diminished subperiosteal bone labeling at eight and 10 weeks indicated that subperiosteal bone remodeling activity had peaked and started to decrease. (**e**–**g**) Chronology of change of the healing activity at different zones based on grading of the bone labeling. In Zone 1 (**e**), markedly similar bone remodeling activities among the three groups was observed. The initial prominent bone activity was observed near the alveolar bone, implant, and bone graft. From 2 weeks, bone remodeling activity decreased and remained at a low level during the observation period. In Zone 2 (**f**), the activity of the control and DCP/HA groups peaked as early as 2 weeks. Activity in the DCP/HA decreased at 8 weeks, earlier than in the control group, which decreased gradually at 10 weeks. In Zone 3 (**g**), the DCP/HA group exhibited higher bone remodeling activity than did the TCP/HA/Col group from 8 weeks, and no obvious bone remodeling activity was observed in Zone 3 during the entire observation period.

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
