# Peer review of "Neutralized Dicalcium Phosphate and Hydroxyapatite Biphasic Bioceramics Promote Bone Regeneration in Critical Peri-Implant Bone Defects"

_materials, 2020, doi:10.3390/ma13040823_

Round 1

Reviewer 1 Report

Neutralized Dicalcium Phosphate and Hydroxyapatite Biphasic Bioceramics Promote Bone  Regeneration in Critical Periimplant Bone Defects

The author fabricated the DCP/HA bioceramics for bone conduction around dental implant. The manuscript needs to be addressed to improve the quality of the article for acceptance.

-In abstract and conclusion section, it is mentioned that neutral pH and optimal compressive strength of DCP/HA bioceramics exhibited good bone formation. However, in this experiment, evidences of these parameters were not evaluated to compare between two types of bioceramics. The author should not mention as conclusion.

-It is better the author add the reference regarding the Resonance Frequency Analysis.

-"Total implant height" abbreviated as TIH in Figure, but TBH in sentence.

-The author select TCP/HA/Col as positive control, however reviewer confuse it. If the author want to mention the advantage of DCP, why did you not contain collagen matrix in DCP/HA?

-I think that TCP/HA/Col have too different point from DCP/HA.

-In fig 4, photographs of histologic specimens have same magnification? Please add the scale burs.

-Fig 4 of downloaded PDF file have no arrows and asterisk.

-How the author histologically detected inflammatory response around the implants?

-In Fig 5, please add the scale bur.

Reviewer 2 Report

Dear authors

The paper looks interesting

Please include into Introduction section those papers 

Ñíguez Sevilla B, Rabadan-Ros R, Alcaraz-Baños M, Martínez Díaz F, Mate Sánchez de Val JE, López-Gónzalez I, Calvo-Guirado JL, De Aza PN, Meseguer-Olmo L. Nurse's A-Phase-Silicocarnotite Ceramic-Bone Tissue Interaction in a Rabbit Tibia Defect Model.J Clin Med. 2019 Oct 17;8(10). pii: E1714. doi: 10.3390/jcm8101714.

Maté-Sánchez de Val JE, Calvo-Guirado JL, Delgado-Ruiz RA, Ramírez-Fernández MP, Negri B, Abboud M, Martínez IM, de Aza PN.Physical properties, mechanical behavior, and electron microscopy study of a new α-TCP block graft with silicon in an animal model.J Biomed Mater Res A. 2012 Dec;100(12):3446-54.

Did you use a disinfection material for the implant surface before biomaterial placement? Please describe 

In the animal experiments, the authors must include some images of mini-surgical saw and another image with the mini saw in dog´s mandible.

Did you use a template for bone defect creation? Please clarify because the technique was not clear.

How did you create the defect? I want to see an image with the bone graft in place covering the bone defects.

Results section

In figure 4, the results were no so successful with and without biomaterials. Can you please use another one? those pictures do not represent the paper

In figure 5  bone regeneration was not close to the implant. Please clarify this

Discussion section 

The authors must describe their results comparing with other papers. I can see the comparison between them.

Conclusion section

It is too poor, you need to rewrite this section and also the conclusions were not so reliable after the results section.

Reviewer 3 Report

An interesting paper exposed by  Chang et al regarding the use of Calcium phosphate and hydroxiapatite biphasic bioceramics in the treatment of bone defects. 

While the paper is well written (both in general presentation, choice and application of methods, result presentation and discussion) there is a need of clarification in the discussion section, with regard to the possible use of angiogenic promoters in the process.

Considering all these, a minor revision of the paper is required. 

Author Response

An interesting paper exposed by Chang et al regarding the use of Calcium phosphate and hydroxiapatite biphasic bioceramics in the treatment of bone defects.
Response: Thank you for your favorable consideration and encouragement.

While the paper is well written (both in general presentation, choice and application of methods, result presentation and discussion) there is a need of clarification in the discussion section, with regard to the possible use of angiogenic promoters in the process.
Response: Thanks for the reviewer's favorable consideration and encouragement. We have added content with references concerning ceramics by creeping substitution can create new vascular channels; therefore, promote angiogenesis during the healing process as followings:
Creeping substitution compromised of osteoclastic resorption, creation of new vascular channels, and osteoblastic bone formation, which can be observed in the early process of healing[4,5]. In this study, we used the bone labeling technique to localize the most active zone of bone remodeling at different time frames. We observed the healing process from a bottom-up direction at the buccal-lingual (coronary) section. (as in Line 581-582)

Considering all these, a minor revision of the paper is required.
Response: Thanks for the reviewer’s favorable comment. We have checked the comment by points to points. We hope to fulfill the requirement for publication.

Round 2

Reviewer 1 Report

No comment

Author Response

Thanks for the reviewer's favorable consideration and encouragement.

Reviewer 2 Report

Dear Authors

I cant see all the Radiographic Examinations at 4 wk, 8 wk, and 12 weeks.

I need all of them please 

Many thanks
